# Effects of Rainfall and Plant Characteristics on the Spatiotemporal Variation of Soil Moisture in a Black Locust Plantation (*Robinia pseudoacacia*) on the Chinese Loess Plateau

**Wenbin Ding [1],*, Fei Wang [2],* and Kai Jin [3]**

1 Key Laboratory of Eco-Functional Polymer Materials of the Ministry of Education, Key Laboratory of Polymer Materials of Gansu Province, College of Chemistry and Chemical Engineering, Northwest Normal University, Lanzhou 730070, China

2 State Key Laboratory of Soil Erosion and Dryland Farming on the Loess Plateau, Northwest A&F University, Yangling 712100, China

3 College of Resources and Environment, Qingdao Agricultural University, Qingdao 266109, China; jinkai-2014@outlook.com

* Correspondence: dwenbin2021@nwnu.edu.cn (W.D.); wafe@ms.iswc.ac.cn (F.W.); Tel.: +86-29-8701-9829 (F.W.)

**Abstract:** Soil moisture is a key factor controlling vegetation construction and ecological restoration in arid and semiarid areas. Understanding its spatiotemporal patterns and influencing factors is essential for effective vegetation water management. In this study, we analyzed the spatiotemporal characteristics of black locust plants using field investigations and statistical analyses and determined the effects of the rainfall and plant characteristics on the soil moisture content (SMC) in a typical watershed in the Loess Plateau, China. The results show that the SMC increases with increasing distance from the tree trunk in the horizontal direction. The vertical profile of the SMC includes layers characterized by rapid decrease, decreased fluctuation, and slow increase. Temporal SMC changes exhibit higher variabilities in the surface layer than in deeper soil layers. Rainfall characteristics notably affect soil moisture. The influence of the rainfall amount is stronger than that of the rainfall duration and intensity. The diameter at breast height, tree height, and canopy width positively affects the soil moisture, whereas the leaf area index and canopy openness negatively affect it. The results of this study provide insights into soil moisture change mechanisms and theoretical references for sustainable plant water use management in arid and semiarid areas.

**Keywords:** rainfall; individual plant; spatiotemporal variation; soil moisture; Loess Plateau

## 1. Introduction

Loess Plateau is a sensitive and fragile ecological zone in which natural vegetation has been severely damaged, and soil erosion has been rampant due to human activities [1,2]. The Chinese government initiated the "Grain for Green" program in 1999 to restore the ecological environment [3]. This program involved massive reforestation efforts, covering an area of $1.6 \times 10^4$ km$^2$, with an investment of USD 8.7 billion, resulting in a 25% increase in vegetation coverage over the past decade [4]. These remarkable reforestation efforts have had positive impacts on the ecological environment, including increased soil carbon sequestration [5], reduced soil loss [6], controlled land desertification, and increased biodiversity [7,8]. However, successful reforestation efforts have also led to several unintended issues, such as intensified soil desiccation and temperature increase in arid and semiarid regions [9,10], which led to a severe conflict between vegetation growth and water resources [11].

The black locust (*Robinia pseudoacacia*), a tree with high drought, infertility, and high/low-temperature tolerance, is considered a major reforestation species in the Loess Plateau [12]. It has been estimated that more than 70,000 hectares of black locust have been planted under the "Grain for Green" program in the Loess Plateau [12]. Black locust

plantations improve the ecological environment in the area by reducing soil erosion, regulating soil respiration, and improving microbiology [13–15]. However, soil moisture in black locust plantations is significantly depleted due to higher transpiration [9], resulting in the formation of a dried soil layer [16], which further aggravates water resource shortages and affects vegetation growth and agricultural production. Thus, studying soil moisture during reforestation processes is vital for water resource management in this region.

Soil moisture is a crucial parameter that regulates the exchange of energy and material in terrestrial ecosystems [17,18]. Spatiotemporal variations of soil moisture affect ecohydrological processes such as infiltration, evaporation, runoff, and plant function and morphology [19], and also affect ecosystem restoration [20], agricultural production, and socioeconomic development [21], particularly in arid and semiarid areas. Therefore, understanding spatiotemporal soil moisture variations and influencing factors is essential for vegetation restoration and improving water resources management in arid and semiarid areas [22].

In recent years, numerous researchers have conducted substantive research on the spatiotemporal variations of soil moisture at different scales. Investigations of the spatial variability in soil moisture have focused on regional and small watershed scales. At the regional level, most regions show significant changes in soil moisture with global temperature rises and changes in rainfall gradients, particularly in the surface soil layer [23,24]. At the watershed scale, the spatial variation of deep soil moisture is affected by the soil depth [25]. Soil moisture generally increases with the soil depth [18]. Temporal stability is a crucial feature of soil moisture, primarily reflecting spatial pattern changes of soil moisture over time [26]. Yang et al. reported that soil moisture is temporally constant at depths ranging from 0.4 to 2.0 m, particularly in exotic vegetation [27].

Factors affecting the spatiotemporal variations of soil moisture are currently among the most debated topics in soil moisture research [28]. Although the spatiotemporal variability of soil moisture depends on many factors, including vegetation, terrain, soil properties, and meteorological characteristics, the dominant factors generally vary by location and scale. For instance, at the watershed scale, land use type and slope position have a greater impact on soil moisture [29]. At the local scale, spatiotemporal variability of soil moisture is primarily determined by precipitation and potential evapotranspiration [30,31], while at the slope scale, topography, and soil texture are the primary factors [32]. Soil moisture at depths ranging from 0 to 10 cm is primarily influenced by the topography factor, whereas the soil moisture at depths ranging from 80 to 100 cm depends on the vegetation type [17]. However, few studies have explored the detailed spatiotemporal variations of soil moisture for individual plants [33], which may hinder the development of soil moisture models and understanding of soil moisture variations during plant growth.

Zhifanggou watershed is a typical region on the Loess Plateau that has experienced a process of serious damage and rapid recovery [34]. The soil moisture variability of a black locust forest has been studied in this watershed [35]. However, the spatiotemporal variations of soil moisture based on individual plants remain lacking, and the effects of rainfall and vegetation characteristics on soil moisture are still poorly understood in this region. Therefore, in this study, nine black locust plants in the Zhifanggou watershed of the Loess Plateau were used to (1) determine the spatiotemporal variation characteristics at the single plant scale and (2) investigate the effects of individual rainfall and plant characteristics on the soil moisture. Our results provide valuable information for forest water resource management and hydrological model modification in arid and semiarid areas.

## 2. Materials and Methods

### 2.1. Study Site

The field experiments were conducted from August 2018 to October 2019 within the Zhifanggou watershed (109°13′03″–109°16′46 E, 36°46′28″–36°46′42″ N) (Figure 1), which is a typical loess hilly-gully area in Ansai County, Shaanxi Province. The Zhifanggou watershed covers an area of ~8.7 km$^2$, encompassing sloping lands and elevations ranging from 0° to 65° and 1010 to 1431 m, respectively [36]. The watershed lies in a region with

a warm, temperate, semiarid climate, with an annual average temperature of 8.8 °C and ~2415.6 h of sunshine each year. The annual evaporation ranges from 1010 to 1400 mm [37], whereas the average annual precipitation is 543 mm and mostly occurs between June and September (70%) [38]. The dominant soil type in the watershed is Huangmian soil, which consists of sand (21% ± 6%), silt (63% ± 3%), and clay (16% ± 4%) [39]. Primary land use categories within the watershed are forestland (21%), shrubland (34%), grassland (18%), and cropland (13%), with black locusts as the main species in forestland [40].

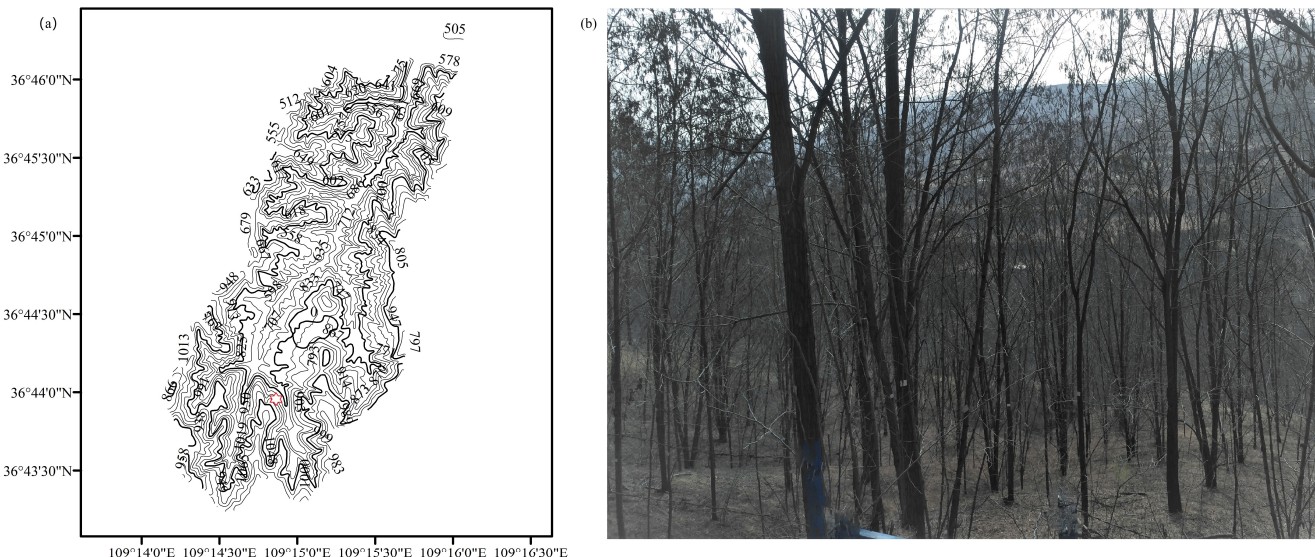

**Figure 1.** (**a**) The topographical map of Zhifanggou watershed and (**b**) the field photograph. Note: The red asterisk represents study site.

## 2.2. Field Experimental Design

The typical black forest site encompasses an area measuring 20 m × 30 m, with a slope of 30° and northwest exposure, situated at an altitude of 1251 m. Based on the results of our field investigation, the average tree height is 9.6 m, with a range of 5.0–15.0 m, and the density of black locust trees in the area is 1600 plants per hectare. The diameters at the breast height (DBH), canopy diameter, and height of trees vary from 6.0 to 23.0 cm, 2.0 to 5.0 m, and 2.5 to 9.0 m, respectively. To ensure the independent and robust nature of the trees utilized in this study, we selected nine plants based on their distribution within the stand area, tree architecture, and canopy structure. These trees are representative of typical black locust trees within the forest and are not affected by surrounding trees. The black locust forest exhibited a limited presence of shrubs and herbs on its surface, and the humus layer was removed prior to the commencement of the experiment. In this study, nine individual black locust plants from the Zhifanggou watershed were used (Table 1).

In this study, several methods were used to measure the tree age, height, DBH, canopy diameter, leaf area index (LAI), and canopy openness [41]. The tree age was determined based on conversations with the forestland owner. The tree height was measured using a tree height meter that could extend and contract up to a maximum of 20 m. The DBH, which refers to the diameter of the trunk at 1.3 m above the ground, was measured using a specialized measuring instrument ending with Zhai [42]. To measure the canopy diameter, a point on the canopy edge was determined through vertical projection to the ground, and a similar point was determined through the trunk center directly opposite to it. This measurement was carried out three times, and the average of the three measurements was used as the result. An unmanned aerial vehicle was used to take photographs of the canopy above the studied trees, which were then imported into the Gap Light Analyzer 2.0 software [43]. The Sidelook 1.1 software was also used to determine the threshold value [44], and this information was then used to calculate LAI in the Gap Light Analyzer

2.0 software. The canopy openness was analyzed using Gap Light Analyzer 2.0 software, which captured the image using a fisheye lens.

**Table 1.** Structural and morphological characteristics of black locust trial trees.

| Tree Code | Tree Age/a | Tree Height/m | Diameter at the Breast/cm | Canopy Diameter/m | Canopy Openness/% | Leaf Area Index |
|-----------|-----------|---------------|---------------------------|-------------------|-------------------|-----------------|
| T-1 |    | 12.1 | 19.0 | 4.4 | 47.2 | 1.3 |
| T-2 |    | 11.5 | 10.5 | 4.1 | 56.6 | 1.0 |
| T-3 |    | 10.7 | 13.5 | 3.2 | 58.8 | 0.7 |
| T-4 |    | 9.8 | 10.1 | 3.7 | 27.0 | 1.8 |
| T-5 | 20 | 11.2 | 10.0 | 3.8 | 49.8 | 1.0 |
| T-6 |    | 9.1 | 8.2 | 2.7 | 33.7 | 1.8 |
| T-7 |    | 10.3 | 8.9 | 3.1 | 74.9 | 0.5 |
| T-8 |    | 9.5 | 10.0 | 3.0 | 53.1 | 0.8 |
| T-9 |    | 7.9 | 9.7 | 2.9 | 45.3 | 0.9 |

*2.3. Field Measurements*

In black locust experimental stands, soil moisture access tubes were placed at equal distances in four directions (0°, 90°, 180°, and 270°), with the trunks of individual plants serving as the centers. According to the canopy width, the distance between the access tubes was 0.8 m, and three access tubes were installed in each direction (Figure 2b). In total, 84 soil moisture access tubes were arranged in accordance with the distribution of individual plants. To measure the soil moisture content (SMC) of the soil profile (volumetric, %), a moisture monitoring tool (TRIME-PICO-BT TDR, IMKO Micromodultechnik GmbH, Ettlingen, Germany) was used to detect data in the 3 m access tubes of TRIME (Figure 2c) [45]. The SMC was measured at intervals of 20 cm down to a depth of 3 m below the surface from August to October 2018, April to October 2019, and in July 2020. Measurements were taken every 10 days. If precipitation occurred, the SMC was measured after it stopped. In total, SMC data were collected 31 times at each of the 84 observation sites during the study period.

Rainfall data were collected using a HoBo-U30-NRC meteorological station (Bourne, MA, USA), which was installed in an open field ~50 m away from the experiment plot. The monitoring frequency of rainfall data was 30 min. Data included the rainfall amount, rainfall duration, and rainfall intensity.

*2.4. Statistical Analysis*

Statistical analysis was carried out using SPSS v. 17.0 (IBM Company), Microsoft Excel 2019 (Microsoft, Redmond, WA, USA), and R software. The spatiotemporal variability of SWC is represented by the standard deviation (SD) and coefficient of variation (CV) of SMC over time ($SD_t$ and $CV_t$) and space ($SD_s$ and $CV_s$), respectively. Correlations among rainfall, plant characteristics, and SMC were analyzed using Pearson's correlation.

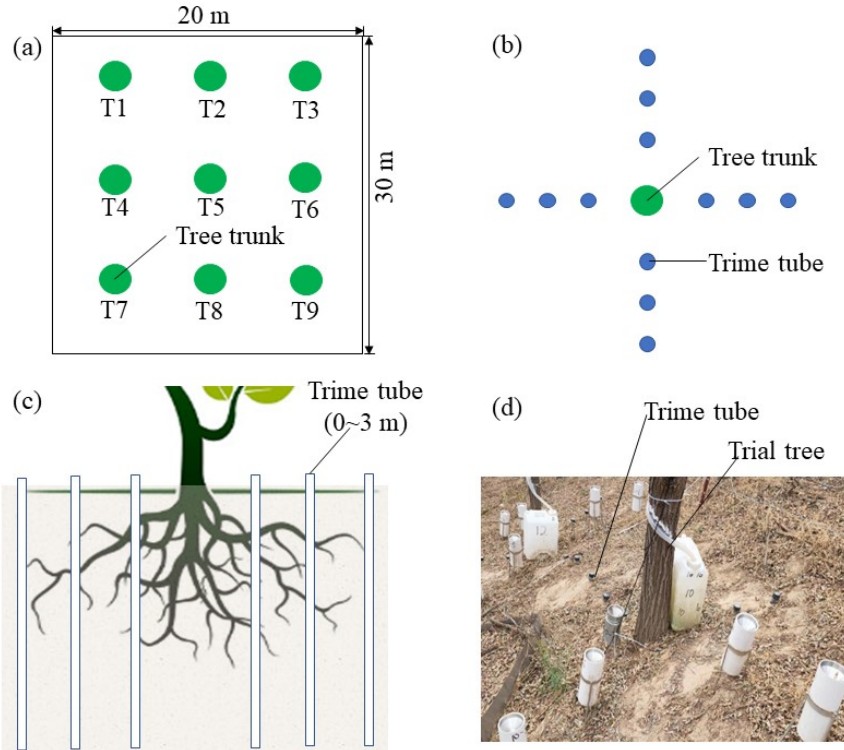

**Figure 2.** (**a**) The locations of the 9 trees selected for the study area; diagrams of soil moisture Trime tubes in the (**b**) horizontal direction and (**c**) vertical profile, and (**d**) field layout photograph.

## 3. Results

### 3.1. Spatiotemporal Variations of the SMC

3.1.1. Spatial Distribution of the SMC

In this study, we analyzed the spatial distribution of trees by utilizing the individual plant trunk as the center and computing the average across all nine trees. The results are shown in Figure 3. Generally, the SMC (averaged across all nine trees) increases with increasing distance from the tree trunk in the horizontal direction (Figure 3a). The average SMC of the 0–300 cm profile during the study period is 7.13%, 7.30%, and 7.67% at distances of 0.8, 1.6, and 2.4 m, respectively. In the vertical direction, the variations in the SMC can be divided into the following three phases (Figure 3b): in the first phase, the SMC rapidly decreases with increasing soil depth (0–80 cm profile); in the second phase, the fluctuation of the SMC decreases (80–180 cm profile); in the third phase, the SMC gradually increases with the soil depth (180–280 cm profile). The vertical distribution of the SMC is consistent with the overall changes at different distances from the tree trunk, whereas the horizontal distribution of the SMC differs from the average value, specifically in the 40–60, 60–80, 160–180, 180–200, and 200–220 cm soil layers (Figure 3c).

3.1.2. Temporal Dynamics of the SMC during the Growing Season

During the 2019 growing season, the SMC was measured 20 times at each of the 84 observation sites. The results are shown in Figure 4. The average SMC of the 0–300 cm profile exhibits an increase in fluctuations throughout the growing season (Figure 4a). During the growing season, the average SMC of the 0–300 cm profile ranges from 4.05% to 14.53%, with the highest and lowest values occurring on 23 May and 4 August, respectively. The highest SMC values were observed in September, whereas the lowest values were observed in May. A slight discrepancy was observed in the variation of the SMC in several soil layers, mainly in the 20–40, 40–60, 60–80, 140–160, and 260–280 cm layers, during the growing season (Figure 4b). In August and September, the SMC was significantly higher

than in other months in different soil layers, indicating that the soil moisture in the deeper soil layer was replenished during this period.

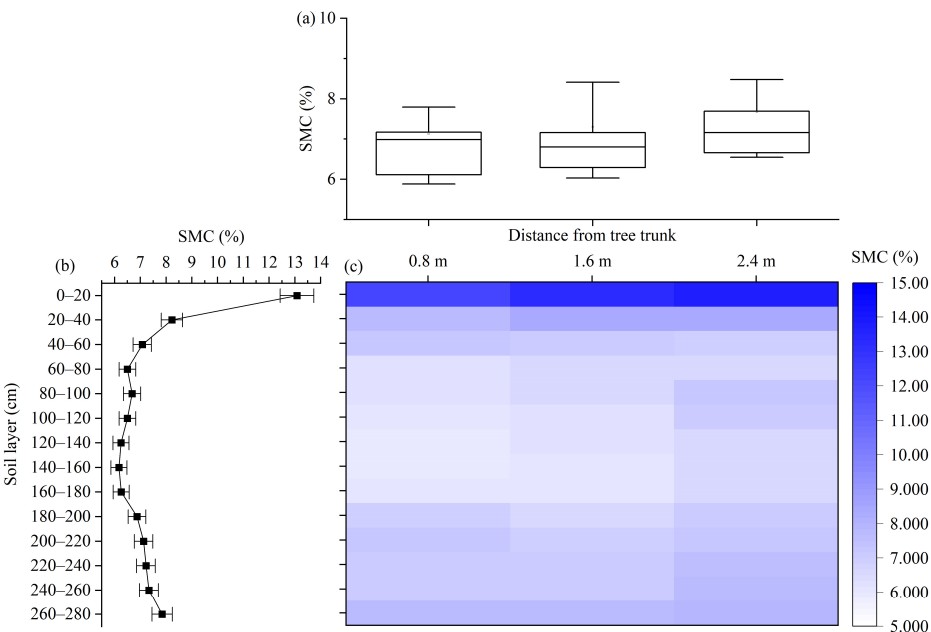

**Figure 3.** Spatial distribution characteristics of the SMC with a single tree as a center. (**a**–**c**) SMC in the horizontal direction, vertical direction, and in different horizontal and vertical locations, respectively.

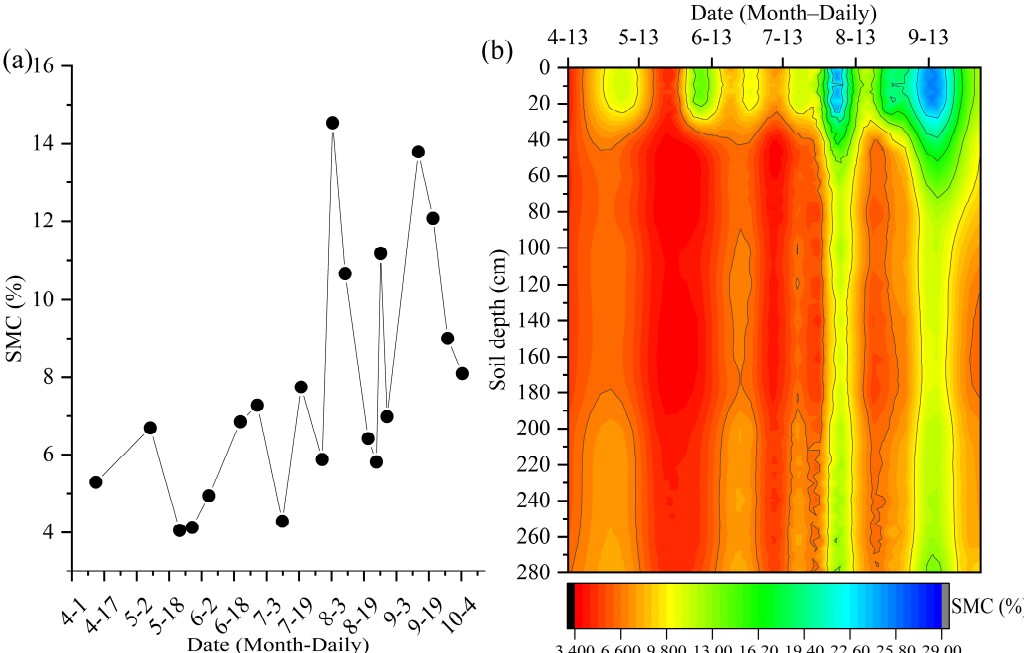

**Figure 4.** Temporal dynamics of the average SMC in the (**a**) 0–300 cm profile and (**b**) at different soil depths during the growing season.

### 3.1.3. Spatiotemporal Variability of the SMC

The coefficient of variation and standard deviation of the same soil layer were analyzed during the growing season (Figure 5). Generally, the $CV_t$ and $SD_t$ show the same patterns of vertical change for different distances from the tree trunk during the growing season. The $CV_t$ exhibits a decrease in the fluctuation with increasing soil depth, with $CV_t$ values ranging from 34% to 57%, all of which belong to the medium variation category. $CV_t$ values

can be divided into three parts; that is, a rapidly decreasing layer at soil depths ranging from 0 to 120 cm and two relatively stable layers at soil depths ranging from 120 to 180 cm and 180 to 280 cm. However, $SD_t$ values can be divided into the following two parts: a rapidly decreasing layer at depths ranging from 0 to 100 cm and a relatively stable layer at soil depths ranging from 100 to 280 cm. The $SD_t$ value ranges from 2 to 8 at different vertical depths for different distances during the growing season.

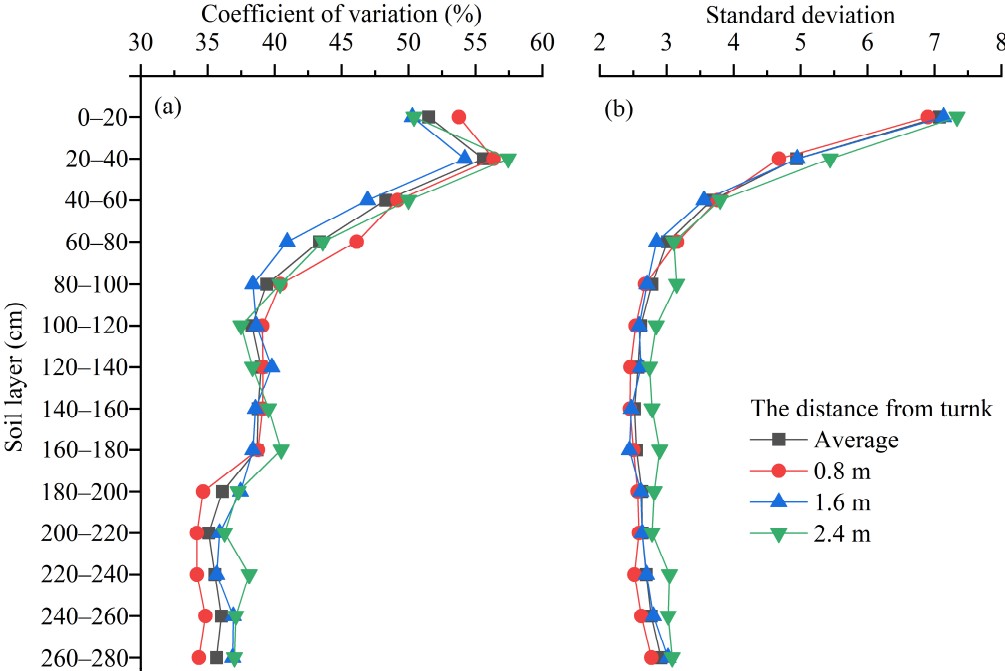

**Figure 5.** (**a**) Coefficient of variation and (**b**) standard deviation of the soil moisture for the same locations during the growing season.

The coefficient of variation and standard deviation were analyzed for different soil depths at the same time, as shown in Figure 6. The $CV_s$ values for soil depths ranging from 0 to 300 cm first increase and then decrease during the growing season. The $CV_s$ values are lower in April–June, September, and October, whereas they are higher in July and August. The $CV_s$ values range from 7% to 52%, with weak and moderate variations. The trends observed for the $SD_s$ variations are consistent with those obtained for the $CV_s$, with $SD_s$ values ranging from 0 to 5.

### 3.2. Analysis of Factors Influencing Soil Moisture

#### 3.2.1. Effects of the Rainfall Characteristics on the SMC

In this study, rainfall and SMC data were collected 25 times. Individual rainfall amounts range from 0.8 to 49.0 mm, with the rainfall duration and rainfall intensity varying from 1.0 to 19.00 h and from 0.46 to 6.20 mm/h, respectively (Figure 7). The data show that there is a similarity between the variations of the SMC (averaged across all 84 observation sites for all depths) and rainfall amount, indicating that the SMC increases with increasing rainfall. However, a similar trend among the rainfall duration, intensity, and SMC could not be observed during the measurement period.

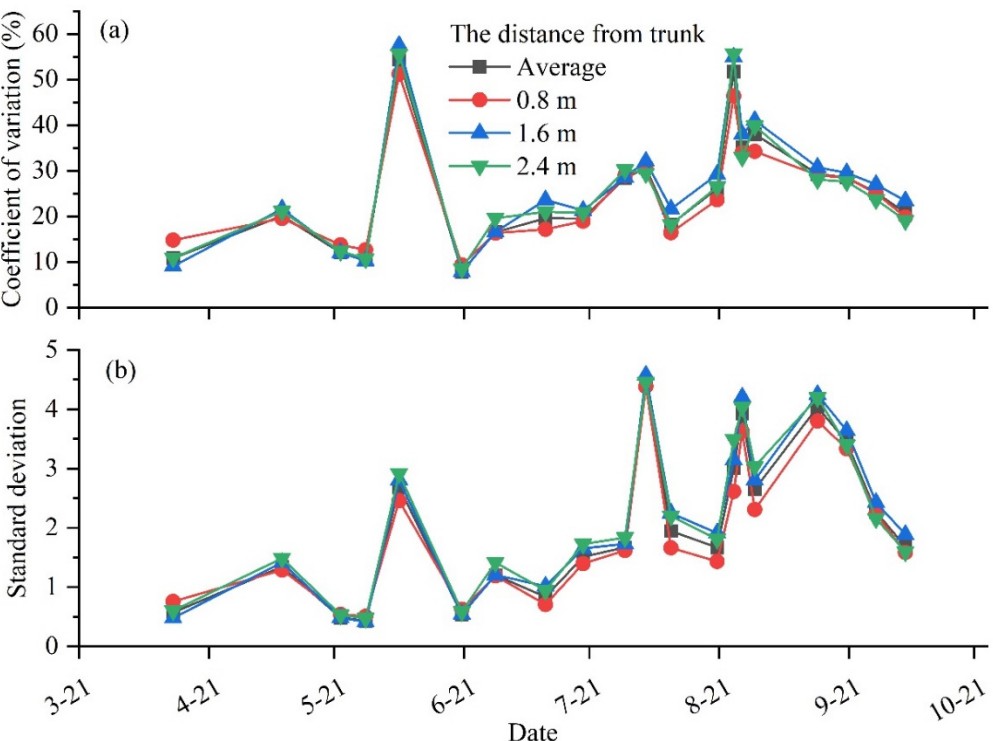

**Figure 6.** (**a**) Coefficient of variation and (**b**) standard deviation of the soil moisture for soil depths ranging from 0 to 280 cm measured during the growing season.

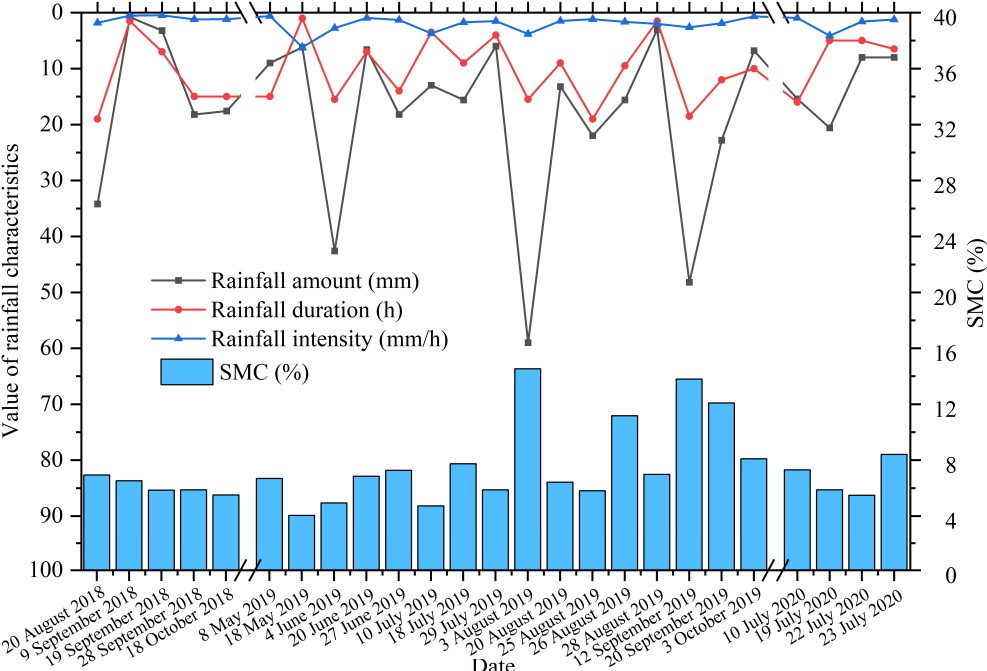

**Figure 7.** Changes of the rainfall characteristics and SMC during the measurement period.

Based on the results of one-dimensional linear regression analysis (Figure 8), the average SMC increases with increasing amount, duration, and intensity of rainfall. Furthermore, the effects of the rainfall amount on the SMC are stronger than those of the rainfall duration and intensity. The coefficient of determination between the average SMC and rainfall amount is higher ($R^2 = 0.34$) than those between the SMC and rainfall duration ($R^2 = 0.13$) and intensity ($R^2 = 0.10$). Correlations among the rainfall amount, rainfall

duration, and SMC are positive in different soil layers, whereas the correlation between the rainfall intensity and SMC is negative, except for the 0–20 cm and 20–40 cm soil layers. The 0–20 cm soil layer yields the strongest correlation coefficient between the rainfall characteristics and SMC. Furthermore, the correlation coefficients between the rainfall amount, rainfall duration, and SMC are significant at the 0.05 level in this layer.

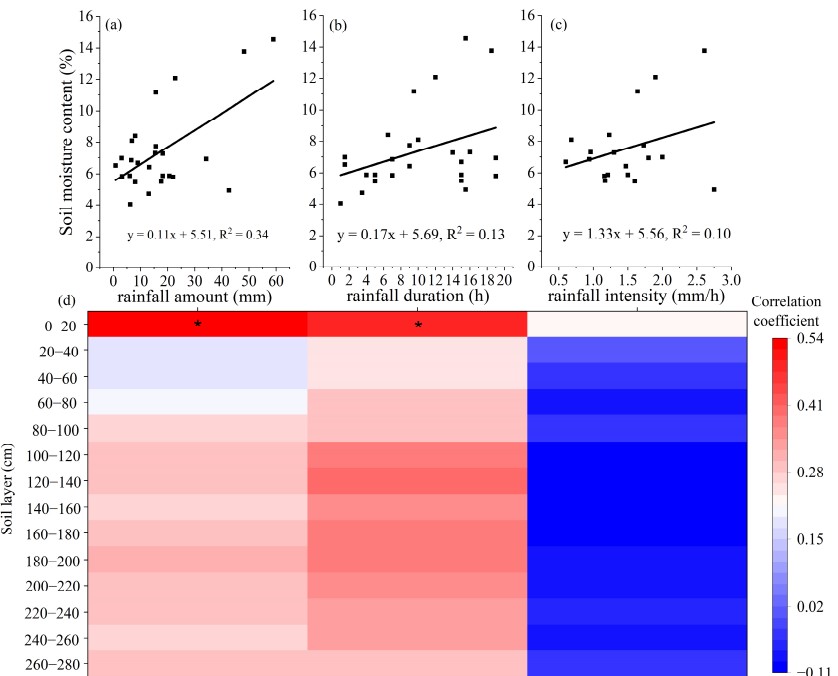

**Figure 8.** Correlations between the rainfall characteristics and average SMC in 0–300 cm profiles (**a**–**c**). Correlation coefficients of the SMC and rainfall characteristics in different soil layers (**d**). Note: Asterisks represent significant correlations at the 0.05 level.

### 3.2.2. Effects of the Plant Characteristics on the SMC

Correlations between the plant characteristics and average SMC in different soil layers are shown in Figure 9. The DBH positively correlates with the SMC in all soil layers except for the 260–280 cm layer; a significant correlation was observed at soil depths ranging from 60 to 180 cm. Similarly, the correlations between the tree height and SMC, as well as the canopy width and SMC, are consistent across different soil layers. The SMC positively correlates with the tree height and canopy width in the soil layer ranging from 0 to 180 cm, whereas a negative correlation was observed in the 180–280 cm layer. However, the correlations among the LAI, canopy openness, and SMC are weak in all soil layers.

Figure 10 presents the correlation coefficient and principal component analysis (PCA) of the effects of different plant characteristics on the soil moisture. The five characteristics were combined into two principal components, which cumulatively accounted for 88.0% of the variance. Specifically, PCI and PCII accounted for 49.5% and 38.5%, respectively. PCI was predominantly influenced by the tree height, canopy width, and DBH, whereas PCII showed a stronger correlation with the canopy openness and LAI. Note that the correlation coefficient between the DBH and SMC (R = 0.78) was the highest and that the correlations between the DBH and canopy width and tree height are better. This suggests that the DBH is a key factor in explaining the PCI variation. Furthermore, the strongest correlation was observed between canopy openness and LAI (R = 0.92), indicating that either variable could explain the PCII variation. Taken together, variations in the SMC are most closely related to the DBH, LAI, and canopy openness.

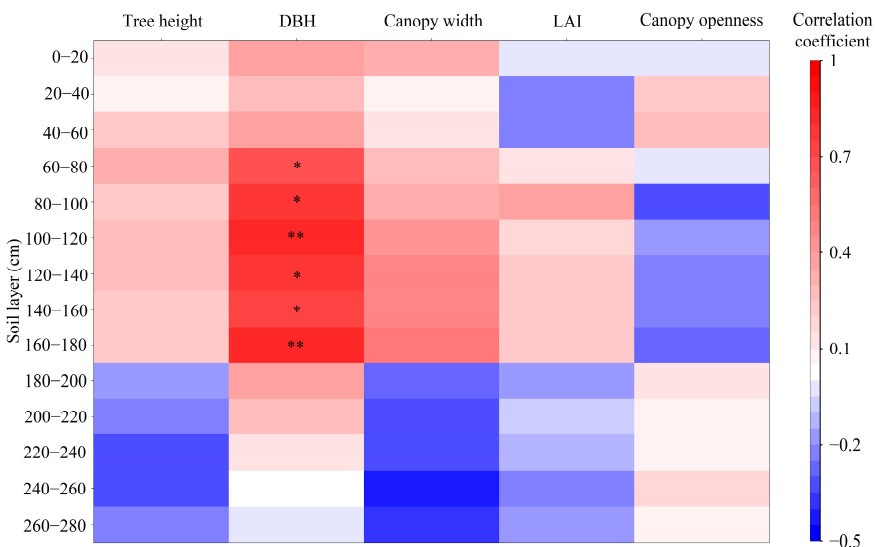

**Figure 9.** Correlation coefficients between SMC and vegetation characteristics in different soil layers. Note: single asterisks and double asterisks represent significant correlations at the 0.05 and 0.01 levels, respectively.

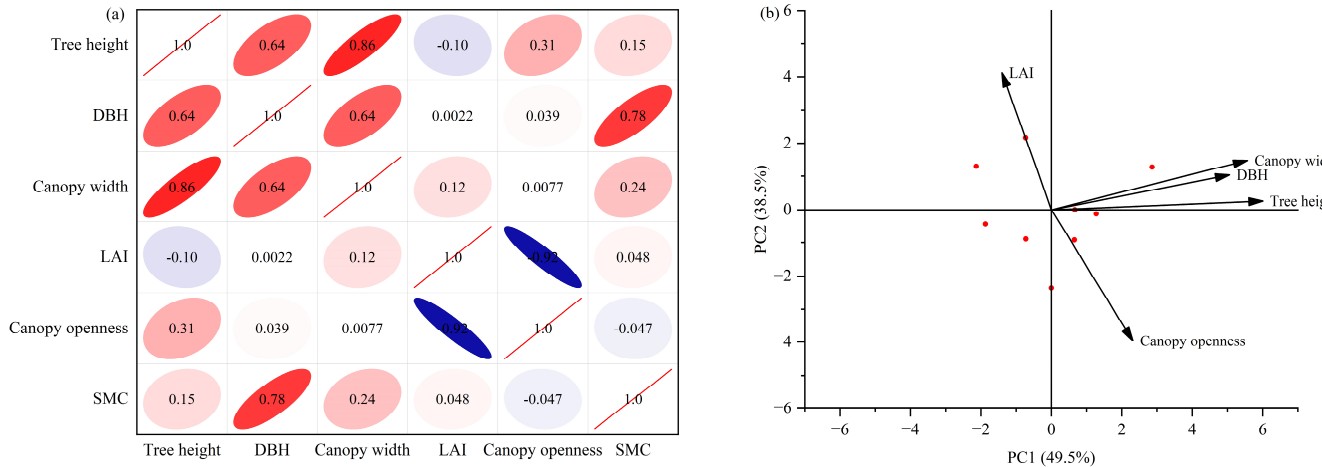

**Figure 10.** (**a**) Correlation coefficient between the plant characteristics and SMC and (**b**)principal component analysis of the effects of the plant characteristics on the SMC.

## 4. Discussion

In this study, we analyzed the spatial characteristics of the SMC in the 0–300 cm vertical profile at varying horizontal distances from the tree trunk. Our results show that the SMC exhibits the following three distinct phases across the soil profile: a rapid decrease with increasing soil depth in the 0–80 cm profile, decreased fluctuation in the 80–180 cm profile, and a gradual increase in the 180–280 cm profile (Figure 3b). This differs from the results of the study by Cheng et al., who reported that the SMC first increases and then gradually decreases and stabilizes in the 0–300 cm profile of black locust forest [46]. This difference may be attributed to rainfall and the plant characteristics. Relate studies indicated that rainfall is a major factor affecting soil moisture in arid and semiarid areas [47]. In our study, the SMC was measured during the rainy season, specifically from April to October. Therefore, the soil moisture could be replenished by rainfall inputs in the soil profile, especially in the surface soil layer. Plant characteristics influence the soil moisture by changing the hydrological processes of rainfall in the soil [48]. Specifically, rainfall intercepted by plant canopies becomes throughfall and stemflow that replenishes the soil moisture; the throughfall distribution can be determined based on the canopy

characteristics [49,50]. In our previous study, the throughfall accounted for 77.71% of the total rainfall, and the horizontal distribution increased with increasing distance from the trunk [41]. This characteristic is consistent with the horizontal distribution of the SMC observed in this study; that is that the SMC increases with increasing distance from the tree trunk (Figure 3a).

During the growing season, the soil moisture in the topsoil layer significantly varies, whereas it remains relatively stable in deeper soil layers (Figure 4). The results of this study show that soil moisture replenishment by rainfall is effective at depths ranging from 0 to 40 cm, with the exception of August and September. During these months, soil moisture is replenished in deeper layers; that is, at depths ranging from 0 to 300 cm. Rainfall, which is concentrated in August and September, can explain this. Wang et al. reported that the rainfall amount predominantly affects SMC variations in arid and semiarid regions [24]. The results of a study on black locust plantations also showed that the SMC consistently increases with increasing rainfall [51], which agrees with our own findings. Furthermore, the fluctuation of the SMC is influenced by both the soil texture and root biomass [52]. Specifically, the soil texture controls the size and arrangement of the soil particles, which in turn affects soil infiltration and evaporation, leading to differences in temporal soil moisture variations [53]. Roots affect the soil moisture in three different ways. Firstly, they affect soil moisture by extracting water from the soil [54]. Secondly, the distribution of roots within the soil profile affects the soil moisture dynamics [55]. Finally, root morphology has an effect on the soil structure, which can impact the soil moisture availability for plant growth [56].

In this study, the focus was placed on the effects of individual rainfall and plant characteristics on soil moisture. Among the various rainfall characteristics, the rainfall amount was determined to be the primary factor affecting soil moisture (Figures 7 and 8), which is consistent with the results reported by Su et al. [57]. Wu et al. stated that rainfall frequencies are more reliable indices of SMC during summer than the rainfall amount, but only non-forest ecosystems yield a better performance [58]. Furthermore, rainfall duration and intensity can affect soil moisture. In general, high-intensity rainfall causes surface runoff and erosion, reducing the amount of water that infiltrates into the soil. However, low-intensity rainfall is more likely to be absorbed by the soil and replenish soil moisture [59]. Short-duration rainfall may not provide enough time for water to infiltrate into deeper soil layers, resulting in limited replenishment of the soil moisture. In contrast, long-duration rainfall can saturate soils and lead to waterlogging [60].

Plant characteristics dramatically affect soil moisture variations by regulating rainfall characteristics [61]. The effects of plant characteristics on the SMC are stronger in the medium soil layer (60–180 cm) than in the surface/subsurface layer (0–60 cm) and deeper layer (180–280 cm; Figure 9). The main reasons are that the soil moisture in the surface/subsurface layers is mainly affected by the rainfall characteristics and that black locust roots are concentrated in the medium soil layer [17,62]. The effects of plant characteristics on soil moisture vary in different regions [63]. The results of our study show that the DBH, tree height, and canopy width have a positive effect on soil moisture. The DBH better reflects soil moisture variation. In contrast, the LAI and canopy openness have a negative effect on soil moisture; both are good indicators for soil moisture variation (Figure 10).

## 5. Conclusions

In this study, the spatiotemporal variation in soil moisture and its determinants were analyzed for black locust plants. The SMC increases with increasing distance from the tree trunk in the horizontal direction. In contrast, changes in the SMC in the vertical direction (0–300 cm) can be divided into the following three processes: rapid decrease (0–80 cm), decreased fluctuation (80–180 cm), and slow increase (180–280 cm). During the growing season, the average SMC of the 0–300 cm profile exhibits increasing fluctuation, whereas slight discrepancies were observed in several soil layers. The temporal variation is higher in surface layers than that in deeper soil layers. The spatial variation is weak to moderate. The ranges first increase and then decrease during the growing season. The SMC is affected

by rainfall and the plant characteristics. The effect of the rainfall amount on the SMC is larger than that of the rainfall duration and intensity. SMC variations are primarily similar to changes in the rainfall amount during the growing season. Plant characteristics mainly affect the soil moisture in the 60–200 cm soil layer. The DBH, tree height, and canopy width positively affect the soil moisture, whereas the LAI and canopy openness negatively affect it. Among these factors, the DBH, LAI, or canopy openness better reflects soil moisture variation. These findings have important implications for the management and conservation of soil moisture in forest ecosystems in arid and semiarid areas.

**Author Contributions:** Conceptualization, W.D. and F.W.; methodology, W.D.; software, W.D.; validation, W.D. and K.J.; formal analysis, W.D.; investigation, W.D.; data curation, W.D.; writing—original draft preparation, W.D.; writing—review and editing, W.D. and K.J.; visualization, W.D.; supervision, F.W.; funding acquisition, W.D. and F.W. All authors have read and agreed to the published version of the manuscript.

**Funding:** This research was funded by the National Natural Science Foundation of China, grant number 42177344, the Gansu Province Science foundation for Youths, grant number 22JR5RA160, and the program to enhance the research capabilities for young teachers of Northwest Normal University, grant number NWNU-LKQN2022-19.

**Data Availability Statement:** Not applicable.

**Acknowledgments:** Ansai soil and water conservation experimental station is gratefully acknowledged for providing experimental site.

**Conflicts of Interest:** The authors declare no conflict of interest.

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
