# Peer review of "Effects of Rainfall and Plant Characteristics on the Spatiotemporal Variation of Soil Moisture in a Black Locust Plantation (Robinia pseudoacacia) on the Chinese Loess Plateau"

_water, doi:10.3390/w15101870_

Round 1

Reviewer 1 Report

This paper focus on the spatiotemporal variation in soil moisture in a black locust (Robinia pseudoacacia) plantation in the Loess Plateau, where black locust is a major reforestation species. This paper therefore provide valuable information for forest water resource management in arid and semiarid region, and the topic is suitable for this journal.

Field experimental design is reasonable, and the results are credible. I recommend to accept this paper after a minor revision. I suggest to make some revisions:

(1) It will be better to add a figure to show study area for 2.1 Study site

(2) Surface herbs, shrubs and humus are also factors that affect soil water, so I suggest to provide some necessary introductions or discussions in the paper.

(3) For the Figures, it will be better to show error amplitude in measurements if you have the repeated measurements. 

Language can make people understand. It's best to do some polishing.

Reviewer 2 Report

Title:

1. Delete “the” from the title as indicated? “Effects of rainfall and plant characteristics on the spatiotemporal variation of the soil moisture in a black locust plantation 3 (Robinia pseudoacacia) on the Chinese Loess Plateau”

Authors and Contact Information:

2.      Are lines 11 and 12 needed?

Introduction:

3.      Line 34: “also” rather than “further”?

4.      Line 34: Is “the” before ecosystem needed? Relevant citation(s)?

5.      Line 35: for socioeconomic development – relevant citation(s)?

6.      Lines 49/50: First sentence of the third paragraph – relevant citation(s)?

7.      Lines 52/53: “the dominant factors generally vary by location and scale” rather than “the dominant factors generally vary among different areas and scales”?

8.      Lines 49-63: In the third paragraph, different scales are mentioned (e.g., small catchment; watershed; local; slope; individual plants) – some rewording to improve clarity in this paragraph would be helpful. For example, do you mean examining, or modelling, or measuring soil moisture at a given scale, etc..

9.      Lines 61-63: please verify the statement that few studies --- or cite those few studies --- what plant(s) or agricultural crop(s) has/have been the focus of the noted few studies?

10.   Lines 79-84: conflicting information is provided: Black locust plantations are said to improve soil water retention but the next sentence then mentions soil moisture is depleted in the plantations due to transpiration.

11.   Lines 90-91: I agree that “Our results provide valuable information for forest water resource management and hydrological model modification in arid and semiarid areas.” but providing more specificity, including citations, would strengthen the comment. Connecting the study focus with the issue of scale (in current 2nd and 3rd paragraphs) would also be helpful for the reader.

12.   I suggest reorganizing and rewording the Introduction (possibly to focus on: understanding soil moisture in Black locust plantations in the Zhifanggou watershed of the Loess Plateau). Possibly begin the introduction with lines 64-86, followed by lines 31-63 + 87-91 including rewording as previously mentioned.

Materials and Methods

13.   Would a figure depicting the study site (the Zhifanggou watershed of the Loess Plateau), its location in the Loess Plateau, including, among others, footprints of the planted Black locust plantations within the Plateau, elevation contours or a DEM for the watershed, County and Province boundary lines, be helpful for the reader?

14.   Lines 103-105: Percent breakdown of land use categories for the study site watershed? A separate zoomed in figure showing their locations within the watershed?

15.   Lines 107-108: Move the lines further below (~Lines 113-117), that summarize tree selection criteria, to be right after this first sentence?

16.   Lines 108-109: I don’t understand this sentence: “The area with typical black locust forest has a size of 20 m × 30 m, is located at an altitude of 1251 m, and has a slope of 30° and northwest exposure.” Please reword for clarity

17.   Another figure showing the locations of the 9 trees selected for the study?

18.   Line 124: citation for the specialized measuring instrument. Reword the sentence to say measured using a … ending with citation(s)

19.   Lines 129-130: citations for the two noted software used for analysis.

20.   Line 135: Is the word strategically needed

21.   Line 137: What was the basis for the determination of the distances between each access tube?

22.   Should references to software and/or measurement devices include a citation (to a user manual for example)

23.   Is the conceptual plot provided in Figure 1 (b) representative of a “typical” root system for a Black locust tree

Results

24.   Line 163: which plant trunk? Average across all nine?

25.   Section 3.1.1 refers to Figure 2, and it is not clear to me what I am looking at. The results in Figure 2 are for which tree among the nine, or are they averaged over all nine trees? What SMC measurement is this from among the 31 that were taken at the 84 observation locations?

26.   Line 180: 20 times at each of the 84 observation sites?

27.   The results in Figure 3 are for which tree and which observation site for that tree? Would it be instructive to modify Figure 3(a) to also include a rainfall hyetograph on the top of the figure

28.   Correct to assume no irrigation?

29.   Change Figure 6 x-axis to show date instead? What is the SMC in this plot? Averaged across all 84 observation sites, for all depths? Isn’t rainfall intensity times rainfall duration equal to rainfall amount?

Returning to the final sentence of the Intro., it would be interesting to read in the discussion and/or conclusions (and abstract) how the study results would specifically inform changes for water management in a black locust plantation in the Zhifanggou watershed of the Loess Plateau, and support modifications of hydrologic models of a black locust plantation in the Zhifanggou watershed of the Loess Plateau

NA

Round 2

Reviewer 2 Report

none